# Insights into tea tree oil-mediated transcriptome modulation in *Rosa hybrida*

**Juliana Lopez-Jimenez[1], Diego Giraldo[2], Felipe Cabarcas [1,3], Natalia Pabon-Mora[4], Juan F. Alzate [1,5]***

1 Centro Nacional de Secuenciación Genómica - CNSG, Sede de Investigación Universitaria- SIU, Universidad de Antioquia UdeA, Medellín, Colombia, 2 STK Bio-Ag Technologies, Research & Development Department, Bogotá, Colombia, 3 Facultad de Ingeniería, Universidad de Antioquia UdeA, Medellín, Colombia, 4 Instituto de Biología, Facultad de Ciencias Exactas y Naturales, Universidad de Antioquia UdeA, Medellín, Colombia, 5 Departamento de Microbiología y Parasitología, Facultad de Medicina, Universidad de Antioquia UdeA, Medellín, Colombia

* jfernando.alzate@udea.edu.co

## Abstract

In this study, we evaluated the impact of substituting conventional antifungal treatments with a commercial Tea Tree Oil (TTO) formulation in *Rosa hybrida* crop plants grown under controlled industrial conditions. Using a transcriptomic approach, we analyzed both leaves and petals to assess the molecular responses to TTO application. Our results revealed a pronounced transcriptomic shift in leaves, where 26 genes were significantly upregulated and one was downregulated, whereas petals displayed more subtle changes. The upregulated genes in leaves were enriched in pathways associated with lipid metabolism, cell wall modification, and plant defense, supporting the view that TTO acts as a bio-stimulant by activating stress-response transcriptional programs. In petals, the few upregulated genes included four transcriptional regulators, while the downregulated set encompassed lipase-like enzymes, cytochrome P450s, and a glucoside malonyltransferase. The comparatively diminished response in petals, which are functional specialized in pollination and have a more limited longevity compared to leaves, supports the view that systemic transcriptional adjustments are more evident in vegetative organs. These findings are consistent with previous reports of TTO's ability to modulate plant stress responses and reinforce its potential as a bio-based alternative to synthetic fungicides in sustainable floriculture.

## Introduction

The global fight against phytopathogenic fungi in agriculture is facing an increasingly difficult outlook due to the emergence of multiple antifungal-resistant strains among major pathogenic species [1,2]. The intensive, and at times indiscriminate, use of chemical fungicides has led to the widespread selection of resistance, a trend now observed across diverse agricultural regions [3,4]. In response, farmers have been

---

**Data availability statement:** The RNA-seq datasets generated and analyzed during this study have been deposited in the NCBI Sequence Read Archive (SRA) under BioProject accession number PRJNA1303024. https://www.ncbi.nlm.nih.gov/bioproject/PRJNA1303024.

**Funding:** The author(s) received no specific funding for this work.

**Competing interests:** The authors have declared that no competing interests exist.

compelled to increase the frequency and dosage of fungicide applications, often implementing more aggressive treatment regimes with reduced intervals between applications [5]. This escalating cycle presents a serious challenge for sustainable agriculture, as it leads to higher production costs, greater environmental impact, and the continued selection of resistant strains. Once resistance has emerged, the continued use of the same or intensified conventional antifungal schemes is likely to promote the development of even more resistant fungal populations.

In this context, natural products, particularly those derived from plants, have gained attention as environmentally friendly alternatives to conventional fungicides [6–8]. Among them, the Tea Tree Oil (TTO), extracted from *Melaleuca alternifolia*, stands out as a promising candidate due to its antifungal properties and potential role in activating plant defense responses [9,10].

*Botrytis cinerea*, the causal agent of gray mold, is one of the most economically significant fungal pathogens affecting a wide range of horticultural crops worldwide, including *Rosa hybrida*. In the floriculture industry, *B. cinerea* poses a persistent challenge due to its necrotrophic lifestyle, high adaptability, and ability to survive under diverse environmental conditions. Roses are particularly susceptible to *B. cinerea* infections, which compromise flower quality and postharvest longevity, leading to substantial economic losses [11,12].

Traditional control strategies rely heavily on synthetic fungicides, often applied multiple times per week. However, the excessive use of chemical treatments has raised growing concerns regarding fungicide resistance, environmental impact, and regulatory restrictions on pesticide residues. These limitations have driven the search for sustainable alternatives, including biopesticides and plant-derived compounds capable of activating the plant's innate immune responses while minimizing ecological harm [13–15].

One promising biocontrol agent is Tea Tree Oil (TTO), a formulation based on the botanical extract of *Melaleuca alternifolia*. Now available in various commercial products, TTO exhibits broad-spectrum antifungal properties and has been reported to act as a plant defense elicitor. Although its efficacy has been demonstrated in several crops, the underlying molecular mechanisms through which TTO modulates host immunity, especially in ornamentals such as roses, remain poorly understood [10,16].

In this study, we used a highly susceptible *Rosa hybrida* cultivar as a model system to evaluate the impact of a commercial TTO-based formulation under crop greenhouse conditions. We compared conventional fungicide management with a treatment scheme partially replacing traditional fungicides with TTO applications. Our approach focused on transcriptomic profiling using RNA-seq technology to identify differentially expressed genes (DEGs) in TTO-treated leaf and petal tissues. Furthermore, we aimed to integrate these transcriptional profiles into enriched biological pathways to gain insight into the potential mechanisms modulated by replacing a conventional chemical fungicide program with a more environmentally friendly TTO-based strategy.

## Methodology

### Experimental site and study design

The field phase of this study was conducted at an industrial rose production facility located in Madrid, Cundinamarca (Colombia). The Momentum rose variety was

selected due to its high susceptibility to *Botrytis cinerea*, the causal agent of gray mold, with the aim of maximizing the sensitivity of the plant system to the evaluated treatments.

The experiment was structured comprising production greenhouses, equally distributed into two main treatments treated as experimental units: i) Conventional management (CM): Treatment based on the company's standard phytosanitary program for the control of *B. cinerea*, which includes up to three weekly applications of chemical fungicides from different modes of action. Recommended commercial doses were used for each product applied in this treatment.

ii) Conventional management + Timorex Pro (CM + TP): A Tea tree oil as an emulsifiable concentrated formulation (Timorex Pro®, 24.5 EC W/V; STK Bio-ag Technologies, Petah Tikva, Israel). Treatment based on the same conventional phytosanitary scheme, with the incorporation of Timorex Pro® (a botanical extract of Melaleuca alternifolia) every 15 days, replacing one or two chemical fungicide applications within the program. The applied dose of Timorex Pro was 1.5 L/ha, following the manufacturer's recommendation. Both treatments were applied continuously over three complete production cycles, under commercial management conditions. The effect of the treatments on the disease was assessed using the company's routine phytopathological measurements, which include the systematic recording of incidence and severity of *B. cinerea* on different plant structures and at harvest.

At the end of the experimental period, leaf and flower samples from random plants across the greenhouse from each treatment were collected, preserved, and processed for subsequent molecular analyses, aiming to evaluate the differential expression of genes associated with plant defense, as well as to perform genetic characterization of the *B. cinerea* populations present.

## Tissue collection and RNA extraction

Plant tissues were selected based on visual inspection by technical staff at the production farm, who identified individuals showing symptoms consistent with *Botrytis* fungal infection. Samples were collected and immediately frozen and preserved in dry ice during transport to our laboratory in Medellín. Total RNA was extracted from each sample using a liquid nitrogen-based tissue maceration protocol, followed by the use of the Plant Concert RNA reagent (Thermo Scientific), according to the manufacturer's instructions. A total of 12 plant tissue samples were processed, including both petals and leaves. RIN measurements were performed in Bioanalyzer 2100 instrument (Agilent Technologies). Samples with the highest RNA integrity number (RIN) values were selected for downstream analysis. All but two samples showed RIN values above 7. The lowest RIN values were 6.0 and 6.6, corresponding to one control leaf sample and one TTO-treated leaf sample, respectively. Based on RIN quality, biological triplicates were retained for each condition: control and TTO-treated petals and leaves.

## RNA-seq raw read data

The RNA-seq datasets generated and analyzed during this study have been deposited in the NCBI Sequence Read Archive (SRA) under BioProject accession number PRJNA1303024, BioSample accessions: SAMN50491601, SAMN50491602, SAMN50491603, SAMN50491604, SAMN50491605, SAMN50491606, SAMN50491607, SAMN50491608, SAMN50491609, SAMN50491610, SAMN50491611, SAMN50491612.

## Differential gene expression analysis

Quality control of the raw RNA-seq reads included adapter trimming and quality filtering performed using Cutadapt [17] in paired-end mode, applying a minimum quality threshold of Q30 and discarding reads shorter than 70 bp. Singleton reads were removed to retain only properly paired sequences. The filtered high-quality reads were then aligned to the *Rosa hybrida* reference genome (ENSEMBL RchiOBHm-V2) using the STAR aligner [18], with the corresponding GFF annotation file to guide transcript-aware mapping. The mapping efficiency was high across all samples, with 88.5–91.0% of reads uniquely aligned to the reference genome, ensuring robust coverage for downstream transcriptomic analyses.

Differential gene expression analysis was conducted using the edgeR (v3.42.4) [19] and limma [20] packages in R. Raw count data from RNA-seq experiments were imported and filtered to remove lowly expressed genes using the filterByExpr() function, which adapts thresholds based on library sizes and experimental design. Genes retained for analysis had a minimum count-per-million (CPM) expression in at least two samples. Comparisons were performed between defined experimental groups, and a design matrix was constructed for generalized linear modeling. Normalization of library sizes was applied using the trimmed mean of M-values (TMM) method. Dispersion was estimated, and differential expression was assessed via a likelihood ratio test (LRT) using the glmFit() and glmLRT() functions. Genes with a false discovery rate

(FDR) < 0.05 were considered significantly differentially expressed. Diagnostic plots, including MDS plots, biological coefficient of variation (BCV) plots, smear plots (MA plots), and volcano plots, were generated to assess data quality and visualize differential expression results.

## Differential expression heatmap visualization

To visualize the expression patterns of upregulated genes, we constructed heatmaps from the RNA-seq data of rose leaves and petals. For each tissue, normalized counts of significantly upregulated genes (FDR < 0.05) were retrieved from the differential expression analysis results. Missing or empty gene descriptions were replaced with "NA" to ensure consistent annotation. Expression matrices were generated separately for leaves and petals (control and treated triplicates). Each gene was labeled in the heatmap using a combined identifier consisting of the gene product ID and its functional description. Counts were $log_2$-transformed after the addition of a pseudocount (+1) to stabilize variance across genes with different expression magnitudes. Heatmaps were generated with the *pheatmap* R package, applying row-wise scaling to emphasize relative expression differences across samples. Control and treated replicates were fixed in the x-axis order, while rows (genes) were hierarchically clustered using Euclidean distance and complete linkage. Color gradients were used to represent normalized expression intensities.

## Results

The RNA-seq experiment generated between 21.5 and 28.2 million raw reads per library. After applying quality control filters (Phred score ≥ Q30, read length > 70 bp, and removal of singletons), at least 92.4% of reads were retained across all 12 libraries. Post-filtering and alignment to the *Rosa hybrida* reference genome (ENSEMBL RchiOBHm-V2), between 18 and 23 million high-quality reads were successfully mapped per library, accounting for approximately 95% mapping efficiency, indicating robust data quality and effective genome coverage (S1 Table). The *Rosa hybrida* genome annotation includes 45,157 protein-coding genes and 4,038 non-coding RNAs. Across all libraries, a total of 37,240 genes were detected with at least one mapped read, demonstrating broad transcriptome coverage and supporting the reliability of the expression data for downstream analysis.

Differential gene expression analysis began with a Multidimensional Scaling (MDS) analysis to evaluate sample similarities and clustering patterns among replicates (Fig 1). In leaf samples, a clear clustering pattern was observed, with control and Tea Tree Oil-treated (TTOT) groups forming distinct clusters, indicating a treatment effect (Fig 1, panel A). In contrast, petal samples did not show consistent clustering between control and treated groups, suggesting that the Tea Tree Oil treatment had a more pronounced impact on the leaf transcriptome, whereas its effect on the petal transcriptome was likely more subtle or even negligible (Fig 1, panel B).

Smear plot analysis of differentially expressed genes (DEGs) between control and treated samples confirmed the trend observed in the MDS analysis (Fig 2). Most genes are clustered around the logFC value of 0 on the y-axis, indicating no differential expression. However, some genes are clearly up- or downregulated, as shown by their displacement toward the upper or lower edges of the plot. Red dots indicate genes with statistically significant changes in expression (FDR < 0.05). As expected, more significant DEGs were detected in the leaf comparison (Fig 2, panel

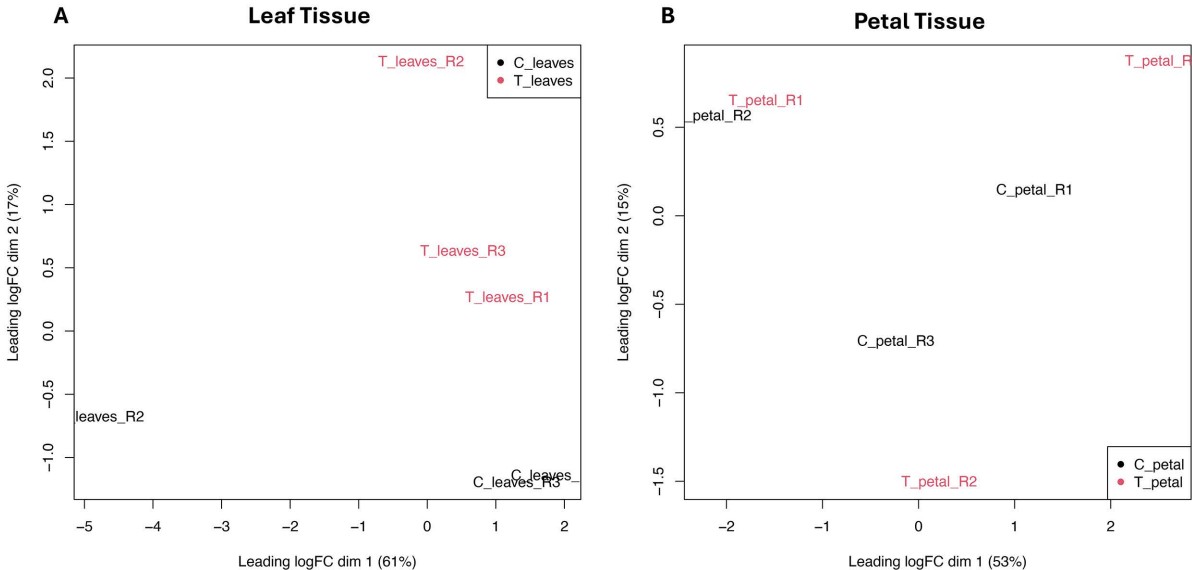

**Fig 1. Multidimensional scaling (MDS) plots of gene expression profiles in rose leaves and petals.** MDS analysis was performed on normalized expression data to visualize the similarity among biological replicates of control and TTO-treated samples. Colors denote experimental groups. **(A)** Leaf tissue. **(B)** Petal tissue. Closer replicates indicate higher similarity in overall expression profiles between samples. The axes indicate the first two dimensions of variation, with percentages in parentheses showing the proportion of variance explained by each dimension.

A) than in the petal comparison (Fig 2, panel B). In petals, relatively few genes showed significant changes between control and the TTOT samples, reflected by the small number of red dots and limited dispersion of logFC values. In contrast, the smear plot for leaves displayed a greater number of significantly differentially expressed genes, with a wider spread in logFC and more red points across a broader range of average logCPM values. These findings are consistent with the MDS results and reinforce the conclusion that Tea Tree Oil treatment induces a stronger transcriptomic response in leaves than in petals.

Similarly, the volcano plot comparing control and TTOT leaves (Fig 3, panel A) revealed a much stronger transcriptomic response. A total of 26 genes were significantly upregulated (FDR < 0.05), all with $\log_2$ fold changes above 1.7; notably, 16 exceeded $\log_2$ fold changes of 3, and six surpassed 5, with the highest reaching 7.7. Only one significantly downregulated gene was observed, with a $\log_2$ fold change of −2.2. These results reinforce the conclusion that Tea Tree Oil elicits a more robust and widespread transcriptional response in leaves than in petals (S2 Table). In comparison, the same plot comparing control and TOTT petals (Fig 3, panel B) further illustrates the limited transcriptomic response in this tissue. Most genes cluster around a $\log_2$ fold change of 0, and only a few show statistically significant differential expression (FDR < 0.05; red dots), indicating a minimal effect of the treatment. Among the few regulated genes in petals, most were downregulated—eight genes exhibited negative $\log_2$ fold changes ranging from −2.9 to −11.9. In contrast, only four genes showed modest upregulation, with $\log_2$ fold changes between 1.4 and 3.4.

### Regulated genes, protein product annotations

We annotated the proteins encoded by the significantly differentially expressed genes (FDR < 0.05) in Rosa leaf tissue treated with Tea Tree Oil extract using the eggNOG-mapper tool and validated the functional annotations with the KEGG database. In leaves, several upregulated genes were associated with enzymes involved in diverse metabolic pathways, whereas in petals, a significant number of putative transcription factors were upregulated (S2 Table).

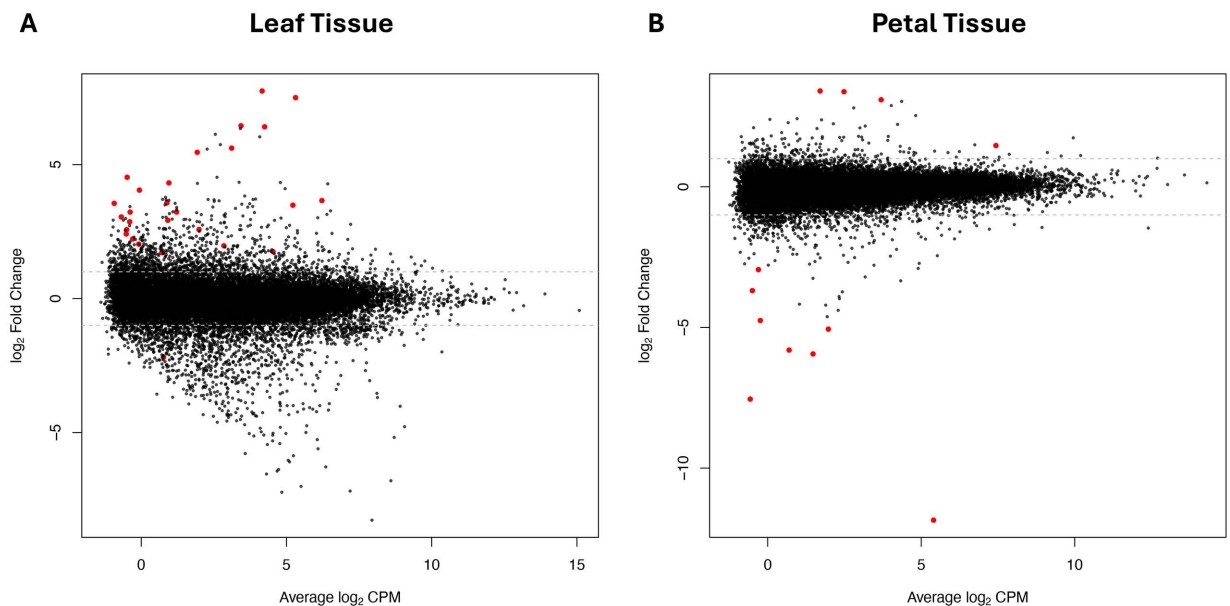

**Fig 2. Smear plot of differential gene expression in *Rosa* leaves.** The smear plot displays log$_2$ fold change (y-axis) versus average log$_2$ counts per million (CPM, x-axis) for all genes analyzed. Each point represents a gene, with significantly differentially expressed genes (FDR < 0.05) shown in red and non-significant genes in black. The dashed horizontal lines at ±1 indicate a fold change threshold. **(A)** Leaves tissue. **(B)** Petal tissue.

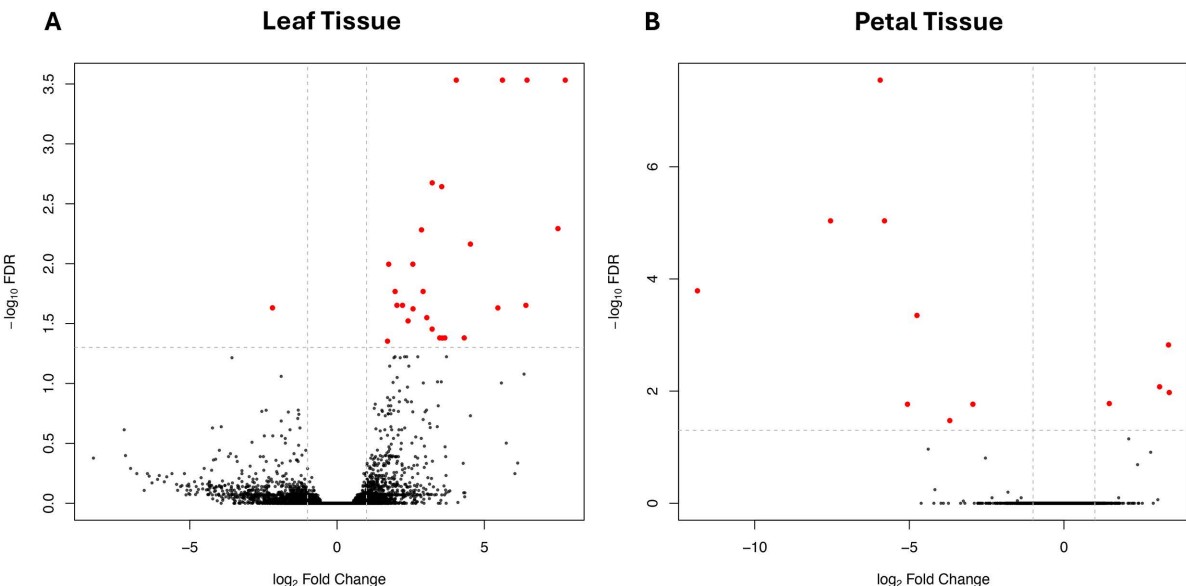

**Fig 3. Volcano plots of differentially expressed genes in Rosa tissues. (A)** Leaves. **(B)** Petals. Each point represents a gene, with the x-axis showing the log$_2$ fold change (treated vs. control) and the y-axis the –log$_{10}$ FDR value. Genes passing the significance threshold (FDR < 0.05) are highlighted in red, while non-significant genes are shown in black. The vertical dashed lines indicate a fold change cutoff (log$_2$FC = ±1), and the horizontal dashed line represents the FDR significance threshold.

## Up-regulated genes on TTOT leaves

Upregulated genes in TTOT leaf samples included several with putative roles in defense, lipid signaling, and cell wall remodeling. The most and third most upregulated genes, PRQ42983 and PRQ42984, were both annotated to KO: K01988, encoding Beta-1,3-galactosyltransferase, an enzyme involved in the glycosphingolipid biosynthesis pathway.

Another highly induced gene, PRQ51976 ($log_2FC = 6.41$), was annotated as a pectinesterase/pectin methylesterase, belonging to the PMR5N/PC-Esterase family. These enzymes modulate pectin de-esterification, altering the structure of the cell wall. We also identified, the gene encoding PRQ18333 ($log_2FC = 4.54$) encodes a Phospholipase. The gene PRQ40556 (logFC = 7.50), the second most highly upregulated in TTOT leaves, is annotated as an At1g04910-like protein, which corresponds to an O-fucosyltransferase (O-FucT). O-FucTs are enzymes involved in protein post-translational modification of proteins (Fig 4).

A particularly noteworthy finding is the upregulation of the gene encoding PRQ54195 (logFC = 4.05), which corresponds to a beta-glucosidase from the Glycoside Hydrolase Family 1 (KEGG: K01188). This enzyme is involved in glycoside hydrolysis. Additionally, the gene encoding PRQ22559 was also strongly induced (logFC = 4.4), which corresponds to a resistance (R) protein and harbours typical domains of this family of defense proteins: LRR3, LRR8, NB-ARC, and TIR.

The gene PRQ51271 (logFC = 3.66), which encodes a protein with a Put_Phosphatase domain and is associated with phosphatase activity, was also significantly induced. Similarly, PRQ48774 (logFC = 1.71), which carries a CYSTM domain typically implicated in infection responses in plants, showed notable upregulation.

Two additional strongly upregulated genes, PRQ53117 (logFC = 4.53) and PRQ44283 (logFC = 3.5), encode proteins annotated as carboxylesterases and harboring Abhydrolase3 domains, suggesting a role in lipid metabolism. In addition, PRQ40482 (logFC = 3.6) encodes an E3 ubiquitin-protein ligase, a central component of the ubiquitin–proteasome system responsible for targeting damaged or misfolded proteins for degradation.

The gene that encodes protein PRQ40778 (logFC = 2.9), a member of the RNase T2 family, was also upregulated. RNase T2 enzymes are involved in RNA degradation and play roles in cellular homeostasis during stress. The gene encoding PRQ26648 (logFC = 2.22) was significantly upregulated and annotated as a calcium-binding protein containing multiple EF-hand domains (EF-hand_1, EF-hand_5, EF-hand_6), which are characteristic of proteins involved in calcium ion sensing and signal transduction. This protein maps to KEGG ortholog K13448 and is associated with the calcium signaling pathway (KEGG: map04626). For the gene product PRQ44084 (logFC = 1.99), bioinformatic annotations retrived that it is related to the HVA22-like protein, a stress-inducible factor implicated in endoplasmic reticulum (ER) stress responses and vesicle trafficking (Fig 4).

Finally, two non-coding RNAs were detected as upregulated in leaves, although no functional annotations were available in the databases used: EPIT00050200742 (logFC = 3.56) and EPIT00050201298 (logFC = 1.97) (Fig 4).

## Down-regulated genes on TTOT leaves

The only downregulated gene encodes a protein annotated in the *Rosa hybrida* genome as PRQ16932, linked to KEGG Orthology term K22418, corresponding to 11β/17β-hydroxysteroid dehydrogenase. Based on eggNOG-mapper functional classification, it belongs to the short-chain dehydrogenase/reductase (SDR) family, a group of NAD(P)(H)-dependent oxidoreductases involved in the metabolism of steroids, lipids, and xenobiotics (Fig 4).

## Up-regulated genes on TTOT petals

In petal tissue, although the transcriptomic response to the Tea Tree Oil treatment was overall more subtle than in leaves, several transcription-related genes were found to be significantly upregulated. Notably, three of the upregulated genes— PRQ58912, PRQ41246, and PRQ16034—encode two-component response regulators. Another more subtle upregulated gene, PRQ58620 (logFC = 1.4), encodes a LHY-like protein (Fig 5).

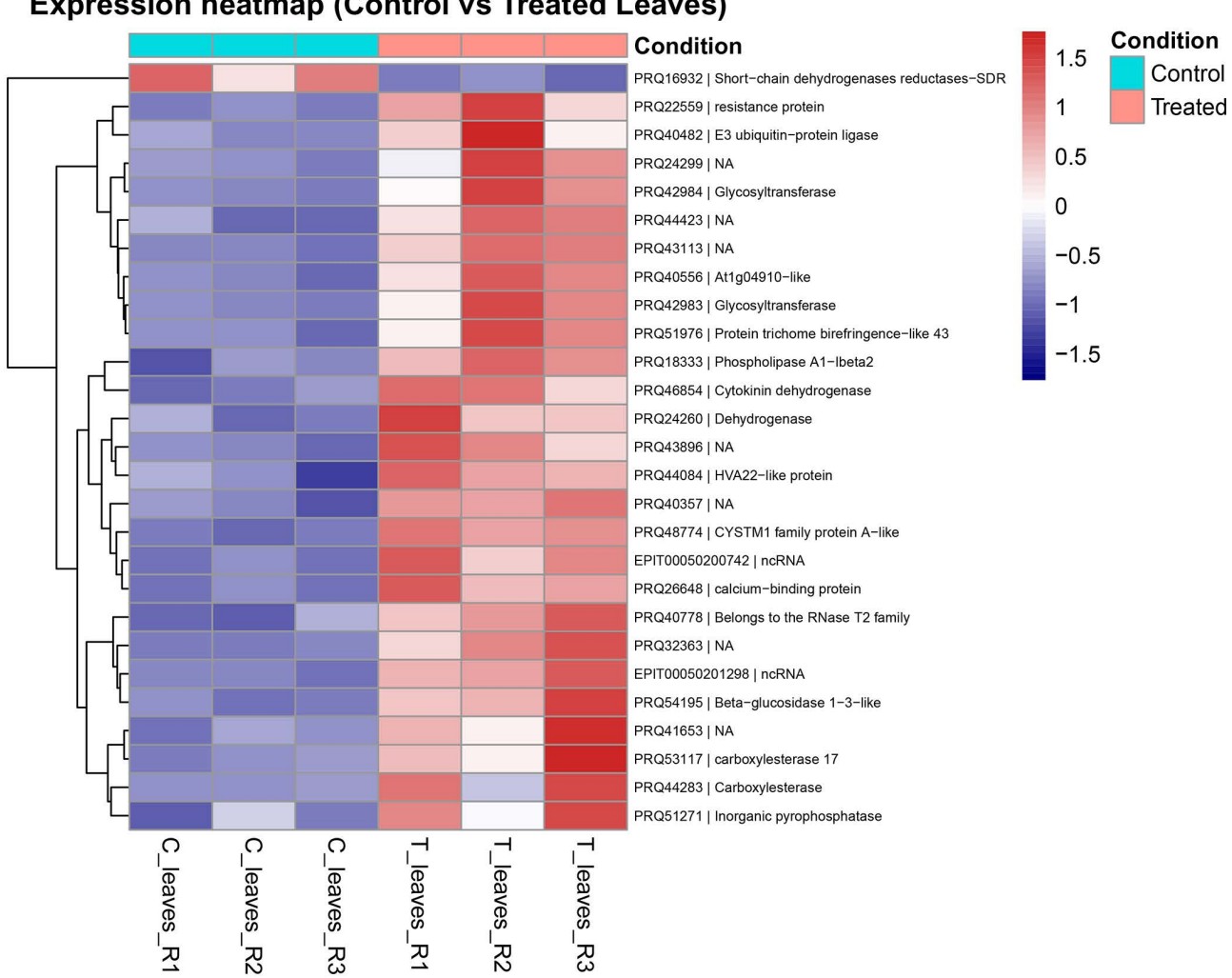

**Fig 4. Heatmaps of regulated genes in rose leaves.** Log$_2$-transformed expression values of significantly upregulated genes (FDR < 0.05) are shown for control and treated leaf tissue (three replicates each). Each row represents a gene labeled with its identifier and functional description, and values are scaled by row to highlight relative changes across conditions. Blue indicates lower expression, white intermediate, and red higher expression. Control samples are shown on the left and treated samples on the right.

## Down-regulated genes on TTOT petals

TTO exposure in petal tissue resulted in the significant downregulation of several genes predominantly associated with specialized metabolism and structural components. Among the most suppressed transcripts was PRQ50438 (logFC = −7.55), annotated as a cytochrome P450 monooxygenase. Another markedly downregulated gene, PRQ54749 (logFC = −3.69), encodes a phenolic glucoside malonyltransferase. Similarly, a cluster of GDSL esterase/lipase-like genes (PRQ47367 and PRQ43725) was consistently repressed, with logFC values ranging from −4.75 to −5.80. Additionally, PRQ25600, encoding a glycine-rich cell wall structural protein 1-like, and PRQ39234, a phylloplanin-like protein from the Pollen_Ole_e_I family, were significantly downregulated (logFC = −5.94 and −11.85, respectively) (Fig 5).

## Expression heatmap (Control vs Treated Petals)

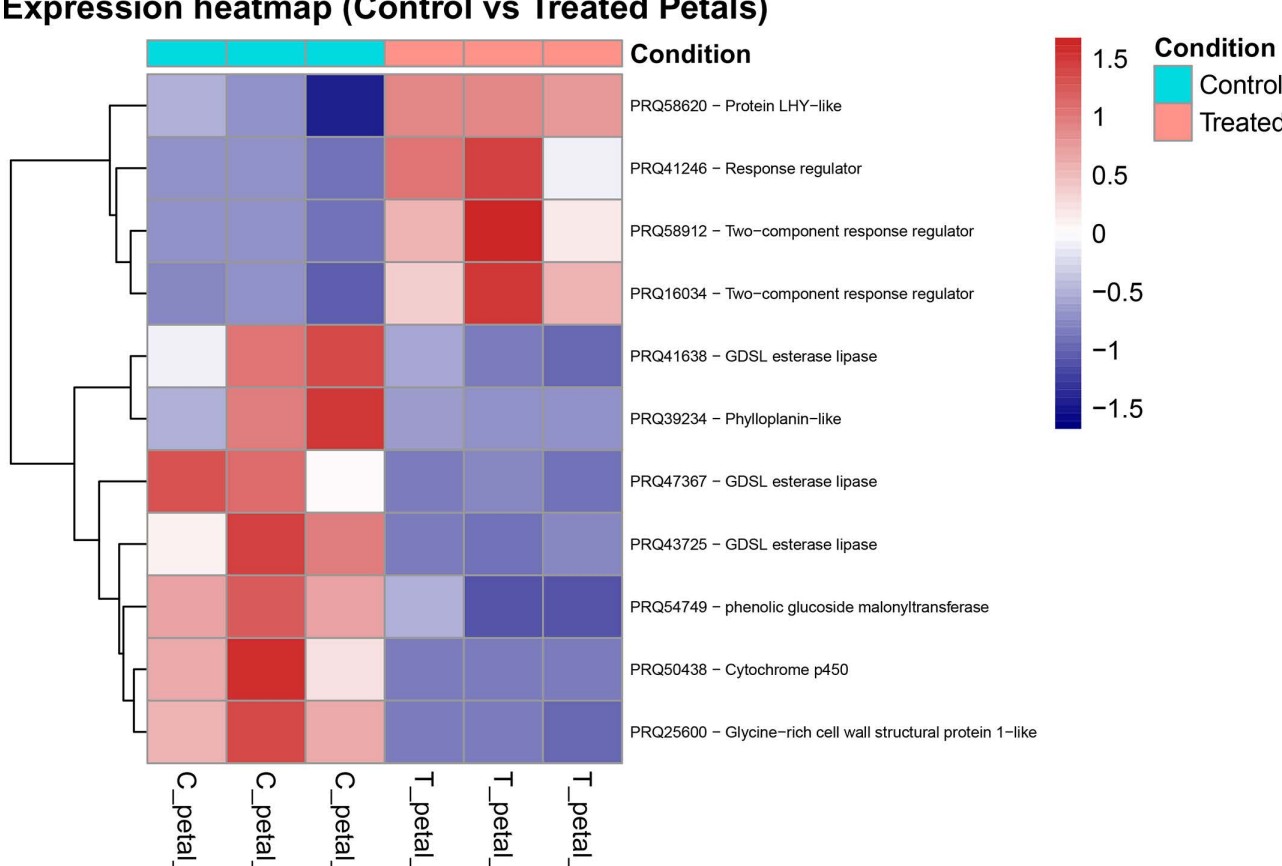

**Fig 5. Heatmaps of upregulated genes in rose petals.** Log$_2$-transformed expression values of significantly upregulated genes (FDR<0.05) are shown for control and treated petal tissue (three replicates each). Each row represents a gene labeled with its identifier and functional description, and values are scaled by row to highlight relative changes across conditions. Blue indicates lower expression, white intermediate, and red higher expression. Control samples are shown on the left and treated samples on the right.

## Discussion

Plant extracts represent a promising innovation for more sustainable agricultural practices, with the potential to replace conventional antifungal chemicals that have been used for decades [21,22]. Compared to traditional antifungal agents, plant extracts offer several advantages, including reduced environmental impact, lower residual effects, and the ability to control highly resistant fungal strains that have emerged due to decades of indiscriminate antifungal use in both developing and developed countries. Additionally, natural plant extracts can act through dual mechanisms in agricultural applications. For example, Tea Tree Oil is well known for its direct antifungal activity [10,23]. Moreover, certain plant secondary metabolites can function as biostimulants, enhancing plant resilience and improving responses to fungal pathogens by priming defense mechanisms [16].

In the present study, we aimed to evaluate the effects of replacing a conventional antifungal treatment scheme in flower plantations with a commercial preparation of Tea Tree Oil (TTO) on *Rosa hybrida* plants used for industrial production. The plants were genetically homogeneous and maintained under identical growth conditions. Using a transcriptomic approach, we investigated potential changes in the plant transcriptome by analyzing both leaf and petal tissues. Our results indicate

that, while the transcriptomic response in petals was minor when comparing the two treatment schemes, a more significant change was observed in leaf tissue. Specifically, TTO treatment led to the differential expression of several genes in leaves, with 26 genes showing significant upregulation and only one gene being downregulated. This supports previous studies suggesting that TTO may act as a biostimulant in plants, eliciting genes involved in stress responses, in this case only in leaves [16].

The upregulated genes in *Rosa* leaves suggest that TTO treatment activates a transcriptional program enriched in enzymes associated with lipid metabolism, cell wall modification, and genes related to disease response and resistance. On the other hand, the subtle response observed in petals is not surprising, as petals are floral short-lived tissues with a highly specialized metabolic and structural role, and likely a more restricted capacity to respond to stress stimuli.

One of the main findings is that two of the most upregulated genes code for enzymes annotated in the glycosphingolipid biosynthesis pathway, specifically encoding Beta-1,3-galactosyltransferases, with expression levels increasing by approximately 90-fold (PRQ42984) and 200-fold (PRQ42983). Glycosphingolipids (GSLs) are not only structural components of the plant plasma membrane but also play critical roles in immune responses, including membrane microdomain organization, signaling, and the regulation of programmed cell death (PCD). Disruption of GSL metabolism has been shown to compromise resistance to both biotrophic and necrotrophic fungal pathogens by affecting salicylic acid levels [24,25]. This pathway is considered a central hub in plant stress response and immunity, mediating both structural defenses and signaling mechanisms. These enzymes catalyze the transfer of galactose residues during the synthesis of complex glycosphingolipids, including members of the lacto-, neolacto-, globo-, and isoglobo-series. Glycosphingolipids are integral components of the plant plasma membrane and have been implicated in membrane microdomain organization, pathogen recognition, and signal transduction [26,27]. The presence of PRQ42983 and PRQ42984 in our dataset suggests an active glycosphingolipid biosynthesis machinery in *Rosa hybrida* tissues, potentially contributing to defense responses against fungal pathogens. Notably, these glycosyltransferases may support the formation of lipid rafts that facilitate the clustering of pattern recognition receptors (PRRs) necessary for effective immune signaling upon detection of fungal PAMPs such as chitin. Additionally, given the known roles of sphingolipid metabolism in regulating PCD, the activity of PRQ42983 and PRQ42984 may influence the outcome of the host-pathogen interaction, particularly in limiting the spread of biotrophic fungi or modulating susceptibility to necrotrophs [24,25,28].

In a concordant manner, genes encoding PRQ51976 (84-fold) and PRQ18333 (23-fold), annotated as pectinesterase/pectin methylesterase and phospholipase, respectively, showed strong upregulation. Pectinesterase/pectin methylesterase belongs to the PMR5N/PC-Esterase family. These enzymes modulate pectin de-esterification, altering the structure of the cell wall. Such changes are commonly associated with abiotic stress responses and pathogen resistance, where pectin remodeling may influence cell wall integrity sensing and defense [29]. Phospholipase is an enzyme that catalyzes the hydrolysis of phospholipids to generate lysophospholipids and free fatty acids—key signaling molecules in jasmonic acid-mediated defense pathways and membrane repair [30,31].

Further supporting the involvement of lipid metabolism in response to TTO exposure is the strong upregulation of two genes encoding carboxylesterases containing an Abhydrolase_3 domain: PRQ53117 (22.6-fold) and PRQ44283 (11-fold). These enzymes are associated with the metabolism of both xenobiotics and endogenous lipid-derived metabolites [32].

Two additional upregulated genes encode products related to the axis of cellular calcium regulation and endoplasmic reticulum (ER) signaling, with the ER being a primary calcium reservoir in plant cells. PRQ26648 (4.7-fold), annotated as a calcium-binding protein containing multiple EF-hand domains, is involved in calcium ion sensing and signal transduction. It plays a crucial role in regulating plant responses to abiotic and biotic stress by modulating intracellular calcium levels and activating downstream stress-responsive genes [33]. PRQ44084 (4-fold) is annotated as a stress-inducible HVA22-like protein related to stress responses in the ER [34].

Another group of positively regulated genes associated with plant defense systems includes PRQ54195 (16×fold change), which encodes a putative β-glucosidase; PRQ22559 (9.2-fold), encoding a putative resistance (R) protein; and

PRQ48774 (3.2-fold), which encodes a CYSTM1 family protein carrying the CYSTM domain, typically present in plant proteins involved in responses to infection. β-glucosidases play a well-established role in carbohydrate metabolism, but more importantly, in plant defense through the activation of phytoanticipins—compounds that are either stored in an inactive form or synthesized de novo in response to pathogen attack [35,36]. These enzymes hydrolyze glycosylated precursors of toxic secondary metabolites, such as cyanogenic glucosides, benzoxazinoids, or coumarins, releasing their active forms upon infection. This activation forms part of the plant's chemical barrier against fungal invasion, particularly effective against necrotrophic pathogens [37,38]. Resistance (R) proteins are well-known components of the plant innate immune system, typically associated with pathogen recognition and immune signaling [39]. The CYSTM protein family belongs to a broader group known as cysteine-rich transmembrane (PCM) proteins, characterized by their cysteine-rich content and localization in the plasma membrane. These proteins are known to be induced upon pathogen exposure and are implicated in plant disease resistance [40].

In addition to genes directly associated with glycosphingolipid metabolism and defense activation, our analysis identified several stress-responsive enzymes with potential roles in mitigating oxidative stress, a common consequence of pathogen attack. Notably, PRQ46854 (K00279), encoding a cytokinin dehydrogenase, and PRQ24260 (K00083), annotated as a cinnamyl-alcohol dehydrogenase, are part of detoxification systems that eliminate toxic aldehydes and alcohols generated through lipid peroxidation and metabolic imbalance during fungal infection [41–44]. These enzymes contribute to the maintenance of cellular redox homeostasis and help limit the damage caused by reactive oxygen species (ROS), which accumulate during the oxidative burst triggered by pathogen recognition. Additionally, PRQ51271 (K06124, K13248), annotated in KEGG as a pyridoxal phosphate phosphatase (PHOSPHO2), may participate in vitamin B6 metabolism—a pathway well-documented for its antioxidant properties in plants. Pyridoxal phosphate, the active form of vitamin B6, can directly scavenge ROS and has been implicated in both abiotic and biotic stress tolerance [45–47].

The co-expression of these redox-related genes, particularly in response to TOTT leaves, suggests that part of the induced defense response may involve the enhancement of antioxidant capacity, aimed at protecting cellular structures during immune activation and minimizing pathogen-induced oxidative damage.

The transcriptomic effect of TTO treatments in petal tissue was markedly different from that observed in leaves, displaying more genes down than up-regulated. Only four genes were upregulated, with logFC values ranging from 1.47 to 3.4. Interestingly, the two most upregulated genes (logFC > 3) were annotated as response regulators, while another encoded a protein annotated as LHY-like. Two-component response regulators are typically involved in hormonal and stress-related signaling pathways, whereas LHY-like proteins are associated with the plant circadian clock and the control of abscisic acid (ABA) biosynthesis and downstream responses [48]. The circadian clock plays a key role in coordinating diurnal gene expression and modulating developmental and stress responses. Moreover, ABA, not only is key for normal germination and growth, as well as drought stress responses, but also is an important signalling molecule activating reactive oxygen species (ROS), which regulate the plant immune system [49]. The upregulation of this gene in TTOT petals suggests that TTO may influence circadian-associated regulatory networks in floral tissues, which warrants further investigation.

These findings suggest that although petals possess a more specialized and limited capacity for environmental responsiveness, TTO treatment is still capable of eliciting transcriptional activation of key regulatory genes involved in signal transduction and temporal control—potentially enhancing the plant's ability to respond to stress during critical reproductive stages.

Conversely, the TTO treatment resulted in the down regulation of several genes, including one encoding a cytochrome P450 monooxygenase (PRQ50438), the most suppressed gene in this tissue. Cytochrome P450 enzymes catalyze diverse reactions in secondary metabolism, especially in phenylpropanoid biosynthesis and detoxification pathways [50]. This repression suggests that TTO might interfere with oxidative metabolism or xenobiotic processing in floral tissues. Similarly, the marked downregulation of PRQ54749, which encodes a phenolic glucoside malonyltransferase, points to a potential TTO-driven alteration of floral pigmentation pathways, as this enzyme is involved in the modification and stabilization of phenolic compounds such as flavonoids.

Lipase-related genes PRQ47367 and PRQ43725, also strongly downregulated, are typically associated with cell wall lipid remodeling, including cutin metabolism and defense-related ester processing [51,52]. Additionally, PRQ25600, annotated as a glycine-rich cell wall structural protein 1-like, and PRQ39234, encoding a phylloplanin-like protein, are both associated with cell wall components [53,54], extracellular matrix architecture and plant–pathogen interactions [55]. Their suppression further reinforces the notion of reduced metabolic investment in structural and defensive functions in petals following TTO treatment, perhaps triggered by more predictable responses in the vegetative portions of the plant.

Collectively, the transcriptional suppression of these genes highlights a consistent downregulation of pathways related to secondary metabolism, lipid remodeling, and cell wall organization. This suggests that TTO may act as a negative regulator of metabolic activity in petal tissues, possibly as part of a stress-adaptive mechanism.

In summary, our results provide evidence that Tea Tree Oil (TTO) treatment can reprogram the transcriptomic landscape of *Rosa hybrida* in a tissue-specific manner, with a more pronounced effect in leaves than in petals. The strong induction of genes involved in glycosphingolipid biosynthesis, lipid metabolism, cell wall remodeling, and redox regulation underscores the dual role of TTO as both an antifungal agent and a biostimulant capable of priming plant defense responses. While floral tissues exhibited a comparatively attenuated response, the upregulation of regulatory genes linked to circadian control and stress signaling suggests that TTO may also influence key processes related to reproductive resilience. Altogether, these findings reinforce the potential of TTO as a sustainable alternative to synthetic fungicides in industrial floriculture, offering both direct antifungal activity and the ability to enhance innate plant immunity through transcriptional modulation. Future studies integrating metabolomic and physiological analyses will be valuable to fully elucidate the mechanistic basis of these responses and to optimize TTO-based strategies for disease management in ornamental crops.

## Supporting information

**S1 Table. Description of the RNA-seq libraries and their statistics.**
(XLSX)

**S2 Table. Up and Down Regulated genes descriptions.**
(XLSX)

## Author contributions

**Conceptualization:** Diego Giraldo, Juan F. Alzate.

**Data curation:** Juliana Lopez-Jimenez, Diego Giraldo, Juan F. Alzate.

**Formal analysis:** Juliana Lopez-Jimenez, Diego Giraldo, Natalia Pabon-Mora, Juan F. Alzate.

**Investigation:** Juliana Lopez-Jimenez, Diego Giraldo, Natalia Pabon-Mora, Juan F. Alzate.

**Methodology:** Juliana Lopez-Jimenez, Diego Giraldo, Natalia Pabon-Mora, Juan F. Alzate.

**Resources:** Diego Giraldo.

**Software:** Juliana Lopez-Jimenez, Felipe Cabarcas, Juan F. Alzate.

**Supervision:** Juan F. Alzate.

**Validation:** Natalia Pabon-Mora, Juan F. Alzate.

**Visualization:** Juliana Lopez-Jimenez, Felipe Cabarcas, Juan F. Alzate.

**Writing – original draft:** Juliana Lopez-Jimenez, Juan F. Alzate.

**Writing – review & editing:** Juliana Lopez-Jimenez, Diego Giraldo, Felipe Cabarcas, Natalia Pabon-Mora, Juan F. Alzate.

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
