## [Decision Letter · Decision Letter 0]

12 Dec 2025

Dear Dr. Alzate,

Thank you for submitting your manuscript to PLOS ONE. After careful consideration, we feel that it has merit but does not fully meet PLOS ONE’s publication criteria as it currently stands. Therefore, we invite you to submit a revised version of the manuscript that addresses the points raised during the review process.

We look forward to receiving your revised manuscript.

Kind regards,

Indu Sharma

Academic Editor

PLOS One

**Journal Requirements:**

3. Please include captions for your Supporting Information files at the end of your manuscript, and update any in-text citations to match accordingly. Please see our Supporting Information guidelines for more information: http://journals.plos.org/plosone/s/supporting-information .

Reviewers' comments:

Reviewer's Responses to Questions

**Comments to the Author**

1. Is the manuscript technically sound, and do the data support the conclusions?

Reviewer #1: Partly

2. Has the statistical analysis been performed appropriately and rigorously?

Reviewer #1: I Don't Know

3. Have the authors made all data underlying the findings in their manuscript fully available?

Reviewer #1: No

4. Is the manuscript presented in an intelligible fashion and written in standard English?

Reviewer #1: Yes

Reviewer #1: Lines "Plant tissues were selected based on visual inspection... showing symptoms consistent with Botrytis fungal infection."This reveals a fundamental flaw: all sampled plants were infected, making it impossible to distinguish:

Direct TTO effects on plant gene expression

Indirect effects mediated through altered disease severity

TTO-pathogen interaction effects.

Confounding treatment with infection status makes it impossible to determine what drives observed expression changes

Absence of disease phenotype data means biological relevance cannot be assessed

Insufficient replication (n=3) limits statistical confidence

Lack of validation leaves findings preliminary

Acknowledge limitations explicitlyi.e insufficient replication, single timepoint, lack of functional validation

**Do you want your identity to be public for this peer review?** For information about this choice, including consent withdrawal, please see our Privacy Policy

Reviewer #1: No

---

## [Author Response · Author response to Decision Letter 1]

22 Dec 2025

PONE-D-25-51544

Insights into Tea Tree Oil-Mediated transcriptome modulation in Rosa hybrida

PLOS One

Journal Requirements:

R:/ Manuscript was adjusted following the Journal style recommendations.

R:/ No specific code was generated for this manuscript. Publicly available software was used, as described in the Methods section.

R:/ Captions for Supplementary tables were added in the revised version of the manuscript.

5. Review Comments to the Author

R:/ We thank the reviewers and the editor for the time invested in the review process and for their thoughtful comments, which have helped us improve the quality of the manuscript.

The RNA-seq raw data generated for this study have been made publicly available in the NCBI Sequence Read Archive (SRA) under BioProject accession number PRJNA1303024, with the following BioSample accessions: SAMN50491601–SAMN50491612.

Reviewer #1: Lines "Plant tissues were selected based on visual inspection... showing symptoms consistent with Botrytis fungal infection."This reveals a fundamental flaw: all sampled plants were infected, making it impossible to distinguish:

Direct TTO effects on plant gene expression

Indirect effects mediated through altered disease severity

TTO-pathogen interaction effects.

Confounding treatment with infection status makes it impossible to determine what drives observed expression changes

Absence of disease phenotype data means biological relevance cannot be assessed

R:/ We appreciate the reviewer’s careful assessment and the opportunity to clarify this point. While Botrytis cinerea is endemic in the commercial production system where the experiment was conducted, disease pressure during the sampling period was very low, and no plants displayed overt or differential disease symptoms. All sampled plants were genetically homogeneous, grown under identical agronomic conditions, and exhibited a similar macroscopic appearance at the time of tissue collection.

Importantly, both control and TTO-treated plants were exposed to the same baseline level of natural Botrytis presence, and no plants with advanced or severe symptoms were selected. Therefore, the only experimental variable distinguishing the two groups was the application of the Tea Tree Oil (TTO) formulation, minimizing confounding effects associated with differential disease severity.

To further address the possibility of pathogen-driven transcriptomic effects, we quantified Botrytis-derived RNA-seq reads across all samples and tissues. These reads were consistently extremely low and comparable between control and treated plants (ranging from n = 39 to n = 289 RNA-seq reads), indicating minimal and relatively uniform fungal biomass across conditions. While we cannot completely exclude indirect effects associated with pathogen presence, these results suggest that the observed gene expression changes are more likely attributable to the introduction of Tea Tree Oil (TTO) rather than to differences in pathogen load or disease progression.

As outlined in the experimental design, our study was not intended to analyze TTO–pathogen interaction mechanisms. We have clarified in the manuscript that pathogen suppression or antifungal efficacy was not part of this work. Furthermore, as noted above, the assessment of Botrytis fungal load revealed very low levels, rendering these RNA-seq data unsuitable for analyzing TTO effects on the fungus. The study was instead designed to evaluate the plant transcriptional response to TTO exposure under realistic field conditions with low disease pressure, reflecting its potential role as a biostimulant or defense-priming agent rather than as a therapeutic antifungal treatment.

Insufficient replication (n=3) limits statistical confidence

Lack of validation leaves findings preliminary

Acknowledge limitations explicitlyi.e insufficient replication, single timepoint, lack of functional validation.

R:/ We thank the reviewer for these comments and respectfully note that RNA-seq–based gene expression analysis is now a mature and widely accepted technology that has largely complemented, and in many contexts replaced, traditional qPCR-based approaches for genome-wide expression profiling. RNA-seq is a well-established method for differential gene expression analysis, and multiple studies have shown that three biological replicates per condition generally represent the minimum sufficient replication to detect statistically robust differentially expressed genes (DEGs) when appropriate analytical methods are applied, especially when working with genetically homogenous organisms maintained under controlled crop conditions (Schurch et al., 2016; Conesa et al., 2016). In our study, we applied a stringent false discovery rate threshold (FDR < 0.05) to control for false positives, supporting the reliability of the observed transcriptional changes. While additional replication and functional validation would further strengthen the conclusions, the current data provide a robust overview, particularly for genes showing stronger regulatory changes, of the plant transcriptional response to TTO under realistic field conditions and constitute a valid basis for hypothesis generation and future targeted studies.

---

## [Editor Report · Decision Letter 1]

13 Jan 2026

Insights into Tea Tree Oil-Mediated transcriptome modulation in Rosa hybrida

PONE-D-25-51544R1

Dear Dr. Alzate,

We’re pleased to inform you that your manuscript has been judged scientifically suitable for publication and will be formally accepted for publication once it meets all outstanding technical requirements.

Kind regards,

Indu Sharma

Academic Editor

PLOS One
---

## [Editor Report · Acceptance letter]

PONE-D-25-51544R1

PLOS One

Dear Dr. Alzate,

I'm pleased to inform you that your manuscript has been deemed suitable for publication in PLOS One. Congratulations! Your manuscript is now being handed over to our production team.

Kind regards,

on behalf of

Dr. Indu Sharma

Academic Editor

PLOS One